# Protocol for evaluation of a virtual wheelchair simulator in assessing mobility skills and cognitive abilities in diverse populations: A multicentric mixed-methods pilot study

Débora Pereira Salgado[1,2]*, Caroline Valentini de Queiroz[2],
Eduardo Lázaro Martins Naves[2], Yuansong Qiao[1], Sheila Fallon[1]

1 Faculty of Engineering & Informatics, Technological University of the Shannon, Athlone, Ireland,
2 Faculty of Electrical Engineering, Federal University of Uberlândia, Uberlândia, Brazil

* A00257244@student.tus.ie, d.psalgado@research.ait.ie, deborapsalgado@ufu.br

## Abstract

### Background

Current wheelchair acquisition, prescription, and training programs often require comprehensive assessments integrating both power mobility skills and cognitive abilities. While wheelchair simulators offer promise for these assessments, but they have not been fully validated.

### Objective

This study aims to develop and refine a protocol for evaluating the feasibility, reliability and preliminary validity of virtual wheelchair simulator metrics in assessing users' current power mobility skills and cognitive abilities, following STARD guidelines. Reference standards include the self-report Wheelchair Skill Test (WST), Power Mobility Road Test (PMRT) and the Montreal Cognitive Assessment (MoCA).

### Methods

This multicentric, mixed-methods pilot study will recruit participants with mobility disabilities, a control group of individuals without disabilities, and healthcare professionals to use a virtual wheelchair simulator. Healthcare professionals will evaluate the simulator's assessments and provide expert feedback on the protocol. Quantitative data will include simulator-derived performance metrics compared to reference standards, and physiological data (e.g., heart rate, skin conductance, temperature, inter-beat-intervals, accelerometer and eye-gaze tracking). Qualitative data (semi-structured interviews) will capture user experiences and insights for protocol refinement. The Quality of Experience (QoE) evaluation framework will assess cognitive workload (NASA-TLX and PAAS), usability (System Usability Scale), immersion

**Data availability statement:** No datasets were generated or analysed during the current study. All relevant data from this study will be made available upon study completion.

**Funding:** The author(s) received no specific funding for this work.

**Competing interests:** The authors have declared that no competing interests exist.

(IGroup Presence Questionnaire), and emotion (Self-Assessment Manikin). Data analysis will include correlation analysis, regression models, thematic analysis, and statistical tests (e.g., independent t-tests, Mann-Whitney U tests) to compare simulator-based performance across groups.

## Discussion

This pilot study seeks to fill critical gaps in current wheelchair training and prescription methods by exploring the use of a virtual simulator to objectively assess both cognitive abilities and power mobility skills. Integrating the QoE assessment framework will provide insights into user interactions, ensuring that the simulator supports tailored training and improve user outcomes in mobility, and safety. Future research may extend this protocol to clinical settings to further evaluate its applicability and effectiveness.

## Introduction

Effective powered wheelchair provision requires a comprehensive approach that includes not only wheelchair prescription and acquisition but also ensuring that users can safely and competently operate their wheelchairs [1]. However, current methods often fail to integrate objective assessments of both power mobility skills and cognitive abilities [2–4]. Assessing cognitive abilities is particularly important for occupational therapists, who need to determine whether a user can safely control the wheelchair and make independent decisions while operating it. A comprehensive evaluation framework is essential for optimizing training programs and ensuring an appropriate fit and safe usage [2,5].

Several tools are available for assessing power mobility skills, each with its strengths and limitations. The Wheelchair Skills Test (WST) [6], evaluates users' ability to perform tasks in a standardized environment, providing insights into the safety and effectiveness of wheelchair use. The Power Mobility Road Test (PMRT) [7], combines elements from the Power Mobility Indoor Driving Assessment (PIDA) [8], Functional Evaluation Rating Scale (FERS) [9], and Power Mobility Functional Evaluation Tasks (PMFET) [10] assessments to evaluate both structured and unstructured tasks, offering a comprehensive measure of power mobility skills. While virtual wheelchair simulator studies have traditionally been used for training, some studies have explored their potential for assessing power mobility skills by aligning simulator-based tasks with mobility skills assessment tools [11–13]. There is growing interest in using this technology to support existing evaluation methods, offering a structured environment to examine users' mobility performance in a controlled setting.

Cognitive abilities also play a crucial role in the effective use of power wheelchairs, influencing users' ability to control the device and make safe, independent decisions [2,14]. The cognitive aspects of functioning, including attention, executive function, and spatial awareness, can significantly impact wheelchair performance and safety [2,4,14]. The relation between cognitive abilities and power mobility is essential for

understanding and addressing the needs of individuals with impairments. Research suggests that cognitive deficits can impair driving performance, leading to increased risks and reduced autonomy [2,4,15]. Therefore, evaluating cognitive abilities in conjunction with power mobility skills is vital for optimized wheelchair prescription and ensuring users can safely and effectively operate their wheelchairs.

In immersive technologies like virtual simulators, the Quality of Experience (QoE) framework is essential for evaluating user interactions and optimizing system design [16–18]. QoE encompasses usability, immersion, emotion, and cognitive workload [19–21]. Usability refers to how easily users can operate the system [22,23]; immersion involves the depth of engagement users experience; and emotion addresses the affective responses elicited by the technology [24]. Cognitive workload, which measures the mental effort required for task completion, plays a critical role in determining how effectively users engage with and learn from the simulator [25–27]. In the context of wheelchair simulators, incorporating QoE principles may support the refinement of assessment protocols, ensuring that both user experience and functional outcomes are considered in mobility evaluation. This approach was demonstrated by Vailland et al. [18], whose study validated simulator-based tasks using QoE measures with regular users, providing insights into usability, presence, and cybersickness. Their findings emphasize how user feedback can enhance simulator design, ensuring effective integration into training and assessment programs while identifying areas for improvement in user experience.

Wheelchair training programs have traditionally focused on physical mobility skills development, often overlooking the critical cognitive aspects that are necessary for safe and effective wheelchair operation [2,28]. Key factors such as assessing cognitive abilities and understanding how cognitive load impacts the learning process are frequently neglected. Similarly, virtual wheelchair simulators predominantly emphasized physical mobility skill enhancement and assessment, with limited integration of cognitive assessments into design and training protocols [29–34]. This oversight may result in insufficient preparation for users, potentially undermining their ability to achieve independent and safe powered wheelchair use. Although simulators have demonstrated value in training by providing realistic scenarios and feedback, they largely focus on enhancing power mobility skills and do not offer a holistic assessment of the user's abilities [9,10,18,27,35–37]. Designing programs that incorporate both physical and cognitive training components is essential to ensure safe and autonomous wheelchair mobility.

## Study aims

The primary aim of this pilot study is to develop and refine a protocol for evaluating the feasibility, reliability, and preliminary validity of a virtual wheelchair simulator as a clinical tool for assessing power mobility skills and cognitive abilities. This study does not aim to fully validate the simulator but instead to establish an evaluation protocol that can guide future validation research. Specifically, this study will:

I. Analyse the simulators' capability to identify performance patterns and cognitive assessment variations across different user profiles, including those with varying levels of experience.

II. Assess participants' Quality of Experience (QoE) by examining cognitive load, usability, immersion and emotion response to optimize the simulator for enhanced assessment and engagement.

III. Collect expert feedback from healthcare professionals to evaluate the protocol's practicality in clinical and rehabilitation settings, aiming to improve wheelchair prescription, acquisition, and assessment processes.

The primary outcome of this pilot study is the feasibility of using simulator-derived metrics for power mobility and cognitive assessments, evaluated through system usability, participant adherence and technical performance. Preliminary validity will be explored through concurrent and construct validity, using correlations with references tools (PMRT, WST and MoCA) and comparisons between user groups. Inter-rater reliability will be assessed through healthcare professionals' scoring consistency using simulator session records.

The findings from this study will inform future validation research by identifying which simulator metrics are most relevant for assessment and refining the evaluation protocol, including task structure and scoring methodology. Additionally, this study will help determine the sample size requirements for a full validation study and address any technical or methodological limitations observed during this pilot phase.

## Materials and methods

This pilot study follows a mixed-methods approach to evaluate the feasibility and reliability of a virtual wheelchair simulator for assessing power mobility skills and cognitive abilities. This involves collecting simulator-based performance metrics, alongside quantitative and qualitative data, to assess end-user experiences and responses through post-test interviews.

Feasibility will be assessed by examining usability, technical performance, and participant adherence. Reliability will be explored through inter-rater agreement, comparing simulator-derived metrics with established clinical assessment tools such as the Power Mobility Road Test (PMRT), Wheelchair Skills Test (WST), and Montreal Cognitive Assessment (MoCA). Since performance may improve with repeated exposure to the simulator, test-retest reliability will not be assessed to avoid the influence of learning effects.

### Wheelchair simulator system

The virtual wheelchair simulator that will be used in the pilot study is EWATS system as per Fig 1 [38]. This system combines two primary components: the wheelchair simulator (Fig 1A) and devices for capturing user response data (Fig 1B), which is controlled using a wheelchair joystick adapted for simulator use on a computer. It collects implicit physiological data through tools such as the smartwatch from Empatica (EmbracePlus), HD1080p Logi Camera integrated with the OpenFace library [39], Mindband and OpenVibe Interface [40]. Additionally, it gathers explicit self-report data and feedback from semi-structured interviews conducted with users interacting with the wheelchair simulator. Together, these data sources provide a comprehensive understanding of user experience and performance.

The simulator was developed through a collaborative process with key stakeholders, including wheelchair users, clinicians, and rehabilitation specialists. This co-design approach ensured the system accurately reflects real-world mobility challenges while assessing both motor skills and cognitive abilities. Wheelchair users provided input on usability and task realism, while clinicians ensured alignment with clinical standards like the, WST [6], PMRT [7] and MoCA [41].

Through several iterations and adaptations, the simulator has undergone significant improvements based on feedback from wheelchair users and a control group of non-wheelchair users [42–45]. For instance, previous refinements informed adjustments to the simulator's tasks and algorithms, enhancing its ability to assess power mobility skills and cognitive abilities. In this pilot study, the focus is on evaluating the feasibility of using simulator-derived metrics to estimate scores for the reference's standards (PWRT, WST and MoCA). The study will explore the extent to which its performance metrics align with these reference standards, informing future refinements in subsequent validation research.

The simulator has been specifically tailored to accommodate different user groups. Wheelchair users interacted with virtual tasks designed to replicate real-world mobility challenges relevant to their daily experiences, while the control group, consisting of non-wheelchair users provided a baseline for evaluating the system's sensitivity. While non-disabled individuals may still face challenges using a powered wheelchair, their inclusion allows for a controlled baseline comparison, helping assess whether the simulator can differentiate performance variations based on prior experience rather than underlying physical or cognitive impairments. A more detailed rationale for selecting this control group, including its role in assessing cognitive influences and mobility performance differentiation, is provided in the group description section.

To further refine the simulator, post-test semi-structured interviews will capture end-user feedback to guide future enhancements. This iterative refinement process ensures that the simulator continues to evolve based on real-world needs, making it a robust and adaptable tool for assessing mobility and cognitive abilities in virtual environments.

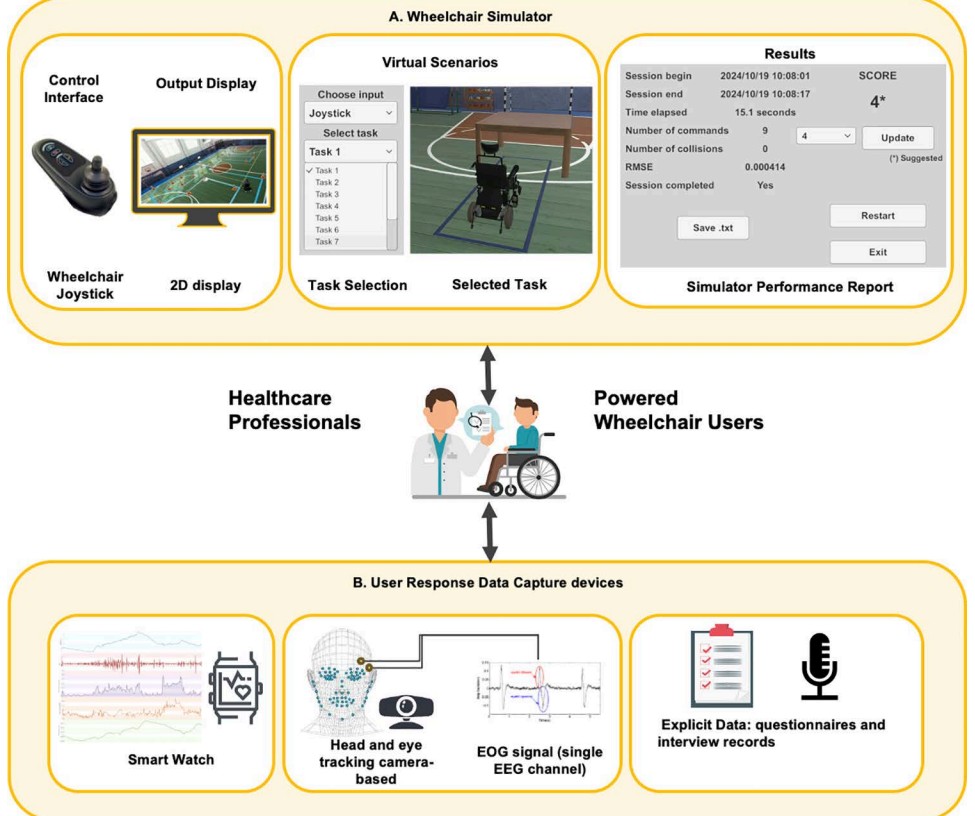

**Fig 1. Wheelchair simulator system.** (A) Diagram of the simulator's core components, including the computer, virtual environment, and control interface (joystick controller). (B) Devices for capturing user responses and physiological data.

## Study design and setting

This pilot study adopts a prospective, multicentric, mixed-methods approach with a cross-sectional exploratory design. The primary aim is to develop and refine a protocol for evaluating the feasibility and reliability of a virtual wheelchair simulator as a clinical tool for assessing power mobility skills and cognitive abilities in wheelchair users with varying levels of experience in controlling a powered wheelchair. Therefore, this study will also explore the simulator's ability to differentiate between novice and experienced wheelchair users while examining the acceptability of the tool for both assessment and training purposes.

This multicentre study will be conducted across multiple locations, including Irish Wheelchair Association (IWA) centres and Technological University of the Shannon (TUS). Conducting the study in multiple settings will provide a diverse participant pool, improving the transferability and generalizability of the findings.

## Participants

The study will encompass three distinct participant groups: wheelchair users, healthcare professionals, and a control group of non-disabled individuals. Participants will be identified through local healthcare networks, TUS and IWA. Data collection will be conducted in two phases.

In Phase 1, wheelchair users will interact with the simulator to assess their power mobility skills and cognitive abilities, with classifications based on their experience level and cognitive function. In Phase 2, healthcare professionals will review

and evaluate the simulator-generated metrics, ranking their clinical relevance and providing feedback on the usability and effectiveness of the tool. The control group, composed of individuals with no prior wheelchair experience, will serve as a baseline to test the simulator's sensitivity in distinguishing variations in mobility and cognitive performance.

In addition to the quantitative and performance data, the study protocol includes semi-structured interviews with people with disabilities and healthcare professionals at the post-test session of each respective phase. These interviews will provide e qualitative insights into the participants' experiences and perceptions of the simulator, contributing to a comprehensive evaluation of its effectiveness and utility.

**Phase 1: Power wheelchair users and control group.** Participants will be power wheelchair users recruited from the IWA and other relevant settings. This phase assesses how effectively the simulator differentiates users based on mobility and cognitive abilities. Wheelchair users will be classified based on their experience level as novice or experienced users and will also be categorized by their cognitive function based on MoCA scores. To improve group comparisons and minimize variability, participants will be stratified according to these classifications. Findings from this phase will guide future validation studies and inform refinements in sampling approaches.

A control group of non-disabled participants with no background in wheelchair control will be included to test the sensitivity of the simulator's algorithms for WST, PMRT and MoCA. This group will serve as a baseline comparison to evaluate the simulator's performance across different cognitive scores and mobility conditions.

**Phase 2: Healthcare professionals.** Healthcare professionals who work with wheelchair users will review the performance data collected from Phase 1. They will examine all the metrics generated by the simulator, rank them in terms of relevance and importance, and validate the scores produced by the simulator. The professionals will also provide feedback on the simulator's utility, usability, and effectiveness in clinical settings. Their input will help ensure the accuracy and clinical relevance of the simulator's assessment metrics.

**Inclusion criteria.** All participants must be aged 18–80 years old and provide written informed consent for study participation and personal data handling and management. Wheelchair users must be able to operate a power wheelchair and engage with the simulator. They will be classified based on experience and cognitive function. Novice users will have limited or no prior experience using a power wheelchair, while experienced users will have at least six months of regular wheelchair use, defined as daily or multiple times per week. Occasional users will not be classified as experienced. Cognitive function will be categorized using MoCA scores: no cognitive impairment for scores 26 and above, mild cognitive impairment for scores between 18 and 25, and moderate cognitive impairment for scores between 10 and 17. Participants with severe cognitive impairment (MoCA below 10) will be excluded due to the potential inability to engage meaningfully with the simulator tasks. Participants must be able to operate a power wheelchair and engage with the simulator. Only manual joystick users will be included to maintain consistency in data collection

Healthcare professionals must have practical experience in wheelchair assessment, prescription, and training. The control group must have no prior experience using power wheelchairs and no known cognitive impairments, with MoCA scores of 26 or higher. Participants will be stratified by experience and cognitive function to improve data robustness.

**Exclusion criteria.** Participants will be excluded if they have severe cognitive impairment, defined as a MoCA score below 10, which would prevent task comprehension and execution. Individuals with severe uncorrected visual or auditory impairments that limit simulator interaction, even with assistive devices, will not be included. Those with severe uncontrolled medical conditions, such as epilepsy with frequent seizures, that pose a safety risk during simulator use will also be excluded. Severe musculoskeletal or neurological conditions preventing upper-limb joystick control also will be a further exclusion criterion. Healthcare professionals without relevant experience in wheelchair assessment or training will not be included. Control group participants with prior powered wheelchair experience or any condition affecting mobility or cognition that could bias simulator performance metrics will also be excluded. The Irish Wheelchair Association (IWA) clinical team will assist in screening participants to ensure eligibility and minimize bias.

**Sampling strategy.** This study employs a stratified sampling approach to ensure diversity in mobility experience and cognitive function, categorizing participants by wheelchair experience (novice vs. experienced) and cognitive function (MoCA scores: no impairment, mild, or moderate cognitive impairment). Their eligibility will be confirmed through initial screening assessments.

The selected approach evaluates the simulator's ability to differentiate mobility performance based on wheelchair experience rather than physical or cognitive impairments. Including non-disabled individuals without wheelchair experience provides a controlled baseline for assessing system sensitivity. By integrating MoCA scores, it assesses how cognitive function interacts with mobility performance, adding another layer to system sensitivity analysis. Defining a control group of wheelchair users without cognitive impairment presents a challenge due to the high variability among wheelchair users in terms of mobility experience and cognitive function. This initial pilot study will help identify an alternative control group for future studies, ensuring a more targeted approach in validation research.

**Sample size.** For this pilot study, a target range of 10–15 individuals per group was selected to ensure feasibility while capturing sufficient variation in mobility experience and cognitive function. If recruitment allows, the sample will be expanded to improve data robustness. This sample size range provides initial representation across skill levels and cognitive abilities, allows for expert feedback from healthcare professionals, and established a baseline for future validation. The selected sample size is consistent with recommendations for pilot and feasibility studies, as outlined by [46,47], balancing between manageability and meaningful data collection goals.

## Assessments

Participants will undergo a series of assessments at different stages: before, during, and after using the simulator as outlined in Table 1. Cognitive functioning will be evaluated using MoCA, while power mobility skills will be measured through the WST and PMRT. Additionally, self-reported questionnaires will assess confidence levels (sub-score of WST) and overall QoE with the simulator.

QoE will be assessed after the simulator experience through several key aspects, including immersive levels (Immersive Experience Questionnaire, IPQ, [48]), emotional responses (Self-Assessment Manikin, SAM, [49]), cognitive workload (PAAS scale [25] after each task and NASA-TLX [50] at the end of test), and usability (System Usability Scale, SUS, [51]).

In Phase 2, healthcare professionals will review the performance records from Phase 1 (wheelchair users and control group), scoring participants' mobility skills using the PMRT. Semi-structured interviews will also be conducted to gather expert feedback on the simulator's impact on both mobility skills and cognitive functioning, as well as its potential for integration into clinical practice.

## Outcomes

The study will evaluate a range of outcomes across primary and secondary categories to assess the effectiveness of the wheelchair simulator, as presented in Table 1. Primary outcomes will include mobility skills measured through the PMRT and WST, cognitive functioning assessed via the MoCA, and evaluations by healthcare professionals during Phase 2, who will review Phase 1 records and score participant performance using the PMRT.

Secondary outcomes will encompass various aspects of QoE, including immersive levels assessed with the IPQ, emotional responses measured with the SAM, and usability evaluated with the SUS, all conducted after the simulator session. Cognitive workload will be assessed using the PAAS after each task and the NASA-TLX following the session. Participants' simulator performance metrics will be collected throughout the sessions. Physiological metrics, including heart rate and stress levels, will be monitored using wearable devices during the tests. This comprehensive evaluation aims to provide valuable insights into the simulator's effectiveness in enhancing mobility skills and cognitive functioning, while also considering the overall experience of participants and the perspectives of healthcare professionals.

**Table 1. Outcomes table.**

| Outcome Category | Measure | Timing | Assessment Method |
|---|---|---|---|
| **Primary Outcomes** | Power Mobility Skills | During the test | Power Mobility Road Test (PMRT) |
| | | Pre- test | Self-reported questionnaire (Wheelchair Skill Test – WST) |
| | | Phase 2: post-test | PMRT scoring by healthcare professionals |
| | Cognitive Functioning | Pre-test | Montreal Cognitive Assessment (MoCA) |
| | Performance Metrics | Throughout Sessions | Collected from the simulator system |
| **Secondary Outcomes** | Quality of Experience | Post-test | Emotional Responses Self-Assessment Manikin (SAM) |
| | | Post-test | Cognitive Load (NASA-TLX) |
| | | During the test | Cognitive Load (PAAS) |
| | | Post-test | Usability – System Usability Scale (SUS) |
| | | During the test | Physiological Metrics (Monitored using wearables devices) |

## Study procedure

The study will follow a structured protocol divided into two phases as per Fig 2, commencing with participant recruitment, which will include wheelchair users, control and healthcare professions participants. Informed consent will be obtained from all participants prior to any assessments and demographics information as detailed in Appendix S1 and S2.

**Phase 1: Wheelchair users and control group.** In the pre-test phase, participants will first rest for five minutes to collect physiological baseline data. Following this, a trained healthcare professional will administer the MoCA assessment to evaluate cognitive functioning. The WST questionnaire will then be completed by participants as a self-reported measure to assess their baseline confidence levels and mobility skills. After these assessments, participants will engage in a simulator free practice session, allowing them to familiarize themselves with the control features and settings of the wheelchair simulator, including speed and acceleration adjustments.

During the test phase, participants will complete a series of predefined tasks based on the PMRT structure tasks while using the simulator. Throughout this simulation, data will be collected on various performance metrics, including task

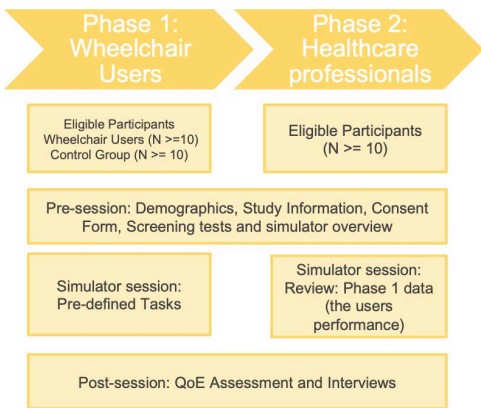

**Fig 2. Study procedure overview.** Schematic representation of the mixed-methods study. The procedure includes participant recruitment, pre-assessment, simulator-based tasks with continuous data recording, post-assessment questionnaires, and final evaluation sessions.

completion time, errors, and joystick control accuracy. Physiological metrics such as heart rate and electrodermal activity will be continuously monitored using wearable devices. Additionally, cognitive workload will be assessed after each task using the PAAS scale.

Following the simulation tasks, participants will undergo a series of post-test assessments. These will include the QoE assessment and open-ended questions to capture qualitative feedback on participants' experiences. Cognitive workload will be measured using the NASA-TLX at the end of the test, alongside the PAAS Scale, which will be administered during the task. The steps of phase 1 are outlined in Fig 3.

**Phase 2: Healthcare professionals.** In this phase, healthcare professionals will evaluate the performance records of wheelchair users and control group from Phase 1. Prior to their assessments, each professional will be briefed on the study's objectives and the specific metrics to consider, and the methodology for evaluation. The professionals will review anonymized simulator session data, including visualization of the wheelchair path and performance records. These observation-based scores will remain confidential and will only be disclosed to the healthcare professionals after they have completed their independent evaluations.

Each healthcare professional will complete the PMRT sheet anonymously, scoring participants' performance based on established criteria, which will evaluate task completion, control proficiency, and overall performance in the simulated environment. This method helps to ensure objectivity and minimizing potential bias. Additionally, they will review results from the Wheelchair Skills Test questionnaire to gain insights into participants' self-reported confidence levels with various wheelchair skills as measured by the simulation.

Healthcare professionals will also complete Quality of Experience assessment and open-ended questions specifically designed to gather qualitative insights into the simulator's potential as a clinical tool, providing valuable feedback on its effectiveness for clinical training and assessment, as well as identifying strengths, limitations, and areas for improvement.

Following their evaluations, healthcare professionals will offer additional qualitative insights based on their observations, contributing to the overall analysis of the simulator's effectiveness in enhancing mobility skills and cognitive functioning. The findings from this phase will be synthesized with the anonymized data from Phase 1 to facilitate a robust analysis of the simulator's impact on participants. The steps of phase 2 are outlined in Fig 4.

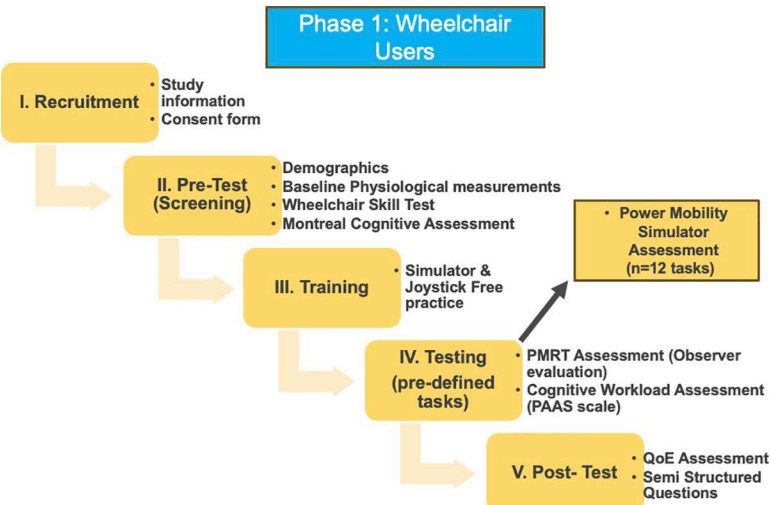

**Fig 3. Phase 1 overview.** Illustration of Phase 1 involving wheelchair users and control participants. This phase includes pre-assessment, simulator task performance, continuous data capture, post-assessment, and comparison of simulator-based metrics with clinical scores.

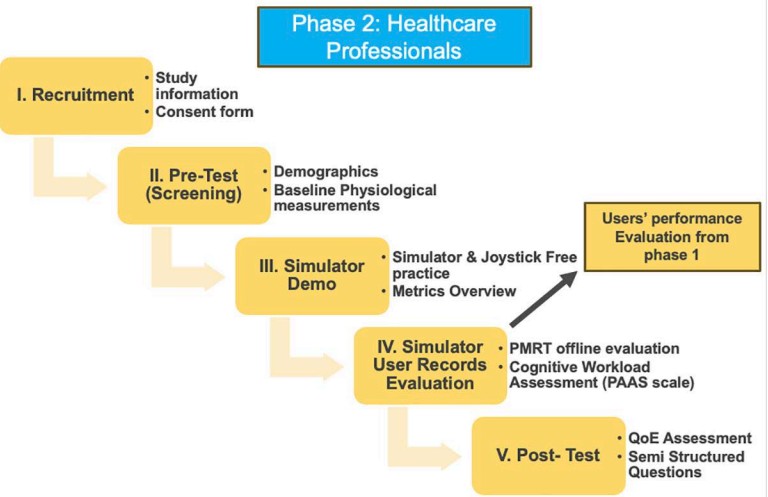

**Fig 4. Phase 2 overview.** Illustration of Phase 2 involving healthcare professionals. Phase 1 performance records are independently reviewed and scored using the Power Mobility Road Test (PMRT) sheet, followed by qualitative interviews and evaluation of the simulator's assessment utility.

### Ethics

This study has been approved by the Research Ethics Committee at Technological University of Shannon (TUS), Athlone, Ireland, before commencing the recruitment. All procedures adhere to the principle of the Helsinki Declaration and follow the STARD (Standards for reporting Diagnostic Accuracy) guidelines [52] for reporting, supplementary material S1 Appendix. Informed, written consent will be obtained from all participants before their involvement in the study. Recruitment has been taken place at the Irish Wheelchair Association (IWA) and TUS facilities from 15th of August 2024–29th May 2025.

### Status and timeline of the study

Participant recruitment began on 15th August 2024 and is currently ongoing, with an estimated end date of 29th May 2025. Table 2 provides an overview of the study's timeline, outlining the stages, their respective status, and detailed descriptions of activities undertaken or planned.

### Test methods

**Index test.** The index test for this study involves the virtual wheelchair simulator, which is designed to assess participants' power mobility skills and cognitive functioning. The simulator presents a series of tasks that replicate real-world challenges, such as navigating ramps and avoiding obstacles. Key performance metrics, including task accuracy, completion time, error rate, and other relevant physiological data, will be logged in real-time by the simulator. The purpose of the index test is to estimate participants' motor skills and cognitive functioning based on the simulator's metrics. These metrics will be compared with scores obtained from the standard assessments.

**Reference standard.** The reference standard consists of three tests: the WST and PMRT for power mobility skills and MoCA for cognitive abilities. They were chosen based on widely used in clinical practice and research, making them suitable for comparison with the virtual simulator's performance [11,53–55]. The WST assessment will be administered before the simulation and PMRT will be administered during the simulation tasks to evaluate changes in mobility skills, while the MoCA will be conducted only as a pre-test to assess participants' cognitive functioning scores prior to the simulation. The MoCA is designed to estimate cognitive functioning across various domains, providing a comprehensive evaluation of cognitive health.

**Table 2. Study timeline.**

| Stage | Timeline | Status | Details |
|---|---|---|---|
| Ethics Approval | 18th December 2023 | Completed | Ethics approval was granted, allowing the study to proceed. |
| System Design Updating and Testing | 15th May 2024–30th July 2024 | Completed | Development and implementation of the user response capture module, testing system functionalities, and ensuring data synchronisation protocols are in place. |
| Phase 1: Recruitment and Consent Form | 15th August 2024–29th May 2025 | Ongoing | Recruitment of participants (control group, wheelchair users) and screening process are ongoing. |
| Phase 2: Recruitment, Consent Form, and Healthcare Professional's Feedback | 18th February – 29th May 2025 (Estimated period) | Pending | After completion of Phase 1 (with a minimum sample of 10 participants), data from Phase 1 (control group and wheelchair users) will be presented to healthcare professionals. |
| Pilot Test – Phase 1 | 9th – 12th December 2024 | Completed | A pilot test was conducted with healthy participants to test the protocol workflow and with experienced power wheelchair users to validate the joystick settings of the simulator. |
| Full Data Collection | 18th February – 29th June 25 (Estimated period) | Pending | Full data collection will involve simulator assessments and post-test feedback. |
| Data Analysis | 1th July – 31st August 2025 (Estimated period) | Pending | After Phase 1 completion and healthcare professionals' feedback in Phase 2, the collected data will be analysed to draw insights and prepare findings. |
| Findings Publication | 1st October 2025 (Estimated period) | Pending | Findings from both Phase 1 and Phase 2 will be compiled and prepared for publication. |

**Test positivity cut-offs.** Simulator-based performance will be categorized based on pre-specified thresholds aligned with PMRT scoring, determined by occupational therapists during the assessment and other healthcare professionals in Phase 2 to ensure clinical relevance. For mobility classification, proficiency will be assessed using task completion time, navigation efficiency (defined as the alignment of simulated trajectory with the optimal path), and error frequency. Task completion time will be measured in seconds, navigation efficiency by mean deviation (cm) and error frequency by number of collisions. Thresholds for these metrics will be established through pilot testing involving individuals from the target population. Receiver Operating Characteristic (ROC) curve analysis will be used to determine the optimal cut-off values for each metric, maximizing sensitivity and specificity.

These thresholds will classify participants into three mobility proficiency categories: proficient, developing, and not proficient categories, mirroring PMRT scores. The proficient category corresponds to a score of 4, indicating completely independent and safe wheelchair operation. The developing category corresponds to a score of 3, representing individuals who complete tasks with hesitation or occasional collisions. A score of 2 indicates that the participant may pose a risk to themselves or others, requiring further evaluation and training. The not proficient category, with a score of 1, signifies an inability to complete tasks, indicating significant mobility limitations and a need for substantial assistance. This classification aims to identify at-risk individuals, assess training effectiveness, and categorize power mobility proficiency.

The WST, in self-report format, will primarily be analyzed as a continuous variable to examine associations with simulator-derived performance metrics. However, exploratory analyses may also investigate the use of a pass/fail threshold or proficiency cut-offs, either based on previous research or determined through pilot data and ROC curve analysis, to assess its potential utility as a classification tool.

Cognitive classification will follow the Montreal Cognitive Assessment (MoCA), categorizing participants as cognitively unimpaired (≥26), mildly impaired (18–25), or moderately impaired (10–17), with those scoring below 10 excluded. Comparative analysis will assess how well simulator-based performance aligns with WST and PMRT scores, identifying potential cut-offs to refine its validity as a clinical assessment tool. Additionally, exploratory cut-offs may be identified through data analysis to explore potential relationships between simulator performance and other variables. However, for reference standards (WST, PMRT and MoCA), established cut-offs based on research and clinical validation will be used, eliminating the need for additional exploratory cut-offs for these standard tests.

**Clinical information access.** To ensure objectivity, assessors conducting the index test and reference standards will have access only to participants' general diagnostic information. Both assessments will be administered under consistent conditions. Offline analysis will be conducted to prevent real-time access to assessment data and maintain assessment rigor.

## Quantitative data analysis

Principal Component Analysis (PCA) will be applied to the simulator' metrics to reduce the dimensionality of the dataset [56,57]. This approach will condense the data into a smaller set of principal components, retaining the essential variance while minimizing redundancy [56,57]. PCA will serve as a preprocessing step for the statistical models (e.g., decision trees, random forests, or support vector machines) by streamlining the input features, thus improving model efficiency and reducing overfitting [58,59]. It will also be used for exploratory data analysis to uncover patterns in the data, such as differentiating or identifying similarities between novice and expert users based on their mobility and cognitive profiles [60,61]. Clustering methods, like k-means, may be applied to these principal components to profile user groups [59,62].

Later, another statistical method will be employed to estimate clinical accuracy metrics, including sensitivity, specificity, and area under the curve (AUC), by comparing the outcomes of the virtual wheelchair simulator with the results of PMRT and the MoCA. ROC curve analysis will be utilized to evaluate the performance of the simulator and determine optimal cut-off points for identifying mobility skills and cognitive abilities. To evaluate the clinical accuracy of the simulator's metrics, the agreement between the estimated scores from the simulator and the actual PMRT and MoCA scores will be assessed using correlation coefficients, Bland-Altman plots, and error measures such as mean absolute error (MAE) and root mean square error (RMSE). Participant age will be recorded and included as a covariate in these statistical analyses. Given that age may influence both mobility performance and cognitive function, adjusted analyses will be performed to control for potential confounding effects and ensure accurate interpretation of simulator-derived metrics. These analyses will assess whether the simulator can reliably support clinical decision-making in mobility and cognitive assessments.

Additionally, artificial intelligence (AI) models will be implemented to estimate PMRT and MoCA scores based on simulator-derived performance metrics. The AI analysis workflow will align with methodologies from peer-reviewed studies [63,64], following a structured approach: first, collecting a range of simulator-derived metrics and applying feature selection to identify the most relevant indicators of driving skills and cognitive abilities. Next, training machine learning (ML) models on datasets where both simulator-derived metrics and actual clinical scores or expert ratings are available.

The primary objective is to predict scores on a 1–4 scale, indicating proficiency in power wheelchair operation, and to estimate MoCA scores for classifying users into cognitive function categories: no impairment, mild impairment, or moderate impairment. Advanced cognitive impairment cases will be excluded as per the study criteria. A range of ML techniques, including regression models, decision trees, random forests, and support vector machines (SVM), will be explored to develop predictive models. These models will be trained using simulator performance data, such as joystick movements, task completion times and navigation accuracy, to establish relationships with standardized clinical assessments. To enhance model accuracy, feature selection techniques, such as PCA will be applied to refine input variables, ensuring the models prioritize the most relevant mobility and cognitive indicators.

Any indeterminate results from the index test will be documented and excluded from the final analyses to maintain the integrity of the clinical accuracy calculations. Furthermore, missing data will be addressed using imputation techniques, such as last observation carried forward (LOCF) or multiple imputation (MI), ensuring a robust analysis while minimizing potential bias in the results [65,66]. LOCF is a single imputation technique commonly used in longitudinal clinical trials to handle dropouts; the subject's last observed value is carried forward to fill in all subsequent missing time points [66]. MI imputation is a modern approach that addresses missing data by creating several complete datasets and combining results [65,66]. Instead of filling in one "best guess" for each missing value, in other words, MI generates multiple plausible values based on the distribution of the observed data and uncertainty about the missing data.

## Qualitative data analysis

The QoE assessments will employ several Likert-style scales tailored to each evaluation instrument. These include the NASA-TLX [50] and PAAS [25] for cognitive load, SAM [49] scale for emotional state, and 5-point scale for usability and effectiveness of simulator [51]. Each instrument's unique scale will be analysed separately to gain quantitative insights into the specific dimensions of experience, such as workload, emotional response, and perceived usability. Descriptive statistics, such as means and standard deviations, will summarize these scales, while further statistical tests may be employed to compare QoE scores across different participant groups.

To complement these Likert-scale data, qualitative data from open-ended feedback provided by healthcare professionals will undergo thematic analysis following the framework described by Braun and Clarke [67,68]. The analysis will begin with transcription and familiarization, ensuring an in-depth understanding of participant responses through repeated reading of the data. Next, initial coding will be conducted by highlighting significant excerpts and grouping them into meaningful categories. These codes will then be examined for patterns and relationships, leading to the development of themes that consolidate related concepts. The final phase will involve reviewing, defining and naming themes, ensuring they represent the perspectives of healthcare professionals regarding the simulator's clinical utility, usability, strength and areas for improvement. This analysis will capture more nuanced insights that are not fully captured by the structured scales, providing a qualitative perspective on the simulator's perceived value and potential limitations in practice.

By integrating thematic findings with quantitative scale-based results will provide a comprehensive evaluation of the simulator's effectiveness in assessing mobility skills and cognitive abilities. Therefore, the combination of quantitative and qualitative insights will inform future refinements and the potential integration of the simulator into clinical assessment and training frameworks.

## Discussion

This pilot study explores the feasibility and reliability of a virtual wheelchair simulator to assess mobility and cognitive abilities in power wheelchair users. By integrating performance metrics with reference standards and user experience data, the study seeks to inform the development of a clinically relevant assessment tool. Participant flow, demographic profiles, and clinical characteristics will be reported to support reproducibility. Cross-tabulations of simulator metrics with WST, PMRT, and MoCA scores will support analysis of clinical accuracy, including sensitivity, specificity, and AUC values. Any adverse events associated with the index or reference tests will also be documented.

A key strength of this study is its integration of mobility and cognitive assessments within a single simulator-based framework. The use of mixed-methods provides a more holistic evaluation by combining clinical benchmarks with physiological data and qualitative feedback. The main limitation is the heterogeneity of the participant group. Wheelchair users vary in condition, experience, and cognitive status, which may limit generalizability. As a small-scale pilot, findings will inform future research but may not yet represent the broader population. Larger and more diverse samples, along with longitudinal follow-up, are needed to evaluate long-term clinical impact. Despite limitations, this study may enhance clinical assessment practices by identifying meaningful simulator-derived indicators. The tool could potentially support not only

initial prescription and assessment but also training and rehabilitation. Insights may also inform future assistive technology development and offer a validated digital framework for combined mobility and cognitive evaluation.

The results of this study will be disseminated through a variety of channels. Publications in peer-reviewed journals focusing on rehabilitation, assistive technology, and cognitive assessment are planned. Findings will also be shared with key stakeholders, including the Irish Wheelchair Association and healthcare professionals, through targeted workshops and presentations. Accessible reports will be made available to clinicians, researchers, and assistive technology developers to foster broader collaboration and innovation.

## Data management, availability and sharing

The de-identified dataset, which includes anonymized participant data, simulator performance metrics, and assessment scores, will be made publicly available for replication and further analysis. All personally identifiable information, such as participant records, images, and other sensitive data, will be securely anonymized, ensuring that no data traceable to individual participants will be accessible. These measures ensure the privacy and confidentiality of study participants.

To adhere to the FAIR principles—Findability, Accessibility, Interoperability, and Reusability—several practices will be followed. Metadata will be created to provide comprehensive information about the dataset, covering data collection methods, key variables, and file formats. The dataset will be stored in a secure, publicly accessible repository on the Open Science Framework (OSF), accessible at https://osf.io/s2tmz/, ensuring ease of access while maintaining data integrity.

The data will be provided in standardized formats, such as CSV and JSON, to facilitate compatibility with various analysis tools and software. Additionally, extensive documentation and metadata will accompany the dataset, enabling researchers to easily understand and reuse the data for future studies, thus contributing to a robust foundation for ongoing research in this field.

## Protocol adaptation and implementation

As the study progresses, amendments may be necessary to adjust participant inclusion criteria, simulator tasks, or assessment protocols based on the feedback received. Any significant changes will be reviewed and approved by the relevant ethics boards. In cases where unexpected challenges—such as technical difficulties or recruitment issues—arise, the study may be paused or terminated. If termination is required, all collected data will be analysed and shared transparently to contribute to the understanding of wheelchair simulators in clinical practice.

The present protocol considers participants' gender, fragility levels, psychological aspects involved in the mental representation of body image, personal endurance, and tolerance toward using wheelchair similar. However, if culture effects emerge, authors will consider implementing a cross-nation design.

## Supporting information

**S1 Appendix. Study information: Includes the procedure checklist, consent form and study overview.**
(DOCX)

**S2 Appendix. Pre-during-post-experience questionnaires.**
(DOCX)

**S3 Appendix. STARD 2015 checklist (Standards for reporting of diagnostic accuracy studies) checklist completed for the article.** N/A – not applicable.
(DOCX)

## Acknowledgments

The authors thank Felipe Roque Martins, Niall Murray and Ronan Flynn for the support provided in the initial steps of the study.

## Author contributions

**Conceptualization:** Débora Pereira Salgado, Caroline Valentini de Queiroz, Eduardo Lázaro Martins Naves, Yuansong Qiao, Sheila Fallon.

**Data curation:** Débora Pereira Salgado, Caroline Valentini de Queiroz.

**Formal analysis:** Débora Pereira Salgado, Caroline Valentini de Queiroz.

**Investigation:** Débora Pereira Salgado, Caroline Valentini de Queiroz.

**Methodology:** Débora Pereira Salgado, Caroline Valentini de Queiroz, Eduardo Lázaro Martins Naves, Yuansong Qiao, Sheila Fallon.

**Supervision:** Eduardo Lázaro Martins Naves, Yuansong Qiao, Sheila Fallon.

**Writing – original draft:** Débora Pereira Salgado, Caroline Valentini de Queiroz, Eduardo Lázaro Martins Naves, Yuansong Qiao, Sheila Fallon.

**Writing – review & editing:** Débora Pereira Salgado, Caroline Valentini de Queiroz, Eduardo Lázaro Martins Naves, Yuansong Qiao, Sheila Fallon.

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
