## [Decision Letter · Decision Letter 0]

7 Feb 2025

PONE-D-24-54281Protocol for Evaluation of a Virtual Wheelchair Simulator in Assessing Mobility Skills and Cognitive Abilities in Diverse Populations: A Multicentric Mixed-Methods Pilot StudyPLOS ONE

Dear Dr. Pereira Salgado, Thank you for submitting your manuscript to PLOS ONE. After careful consideration, we feel that it has merit but does not fully meet PLOS ONE’s publication criteria as it currently stands. Therefore, we invite you to submit a revised version of the manuscript that addresses the points raised during the review process. Congratulations on this initiative! This topic is of great importance to rehabilitation teams. Please see the reviewers' comments and questions below

We look forward to receiving your revised manuscript.

Kind regards,

Rodrigo Rodrigues Gomes Costa, PhD

Academic Editor

PLOS ONE

Journal Requirements:

Reviewers' comments:

Reviewer's Responses to Questions

**Comments to the Author**

1. Does the manuscript provide a valid rationale for the proposed study, with clearly identified and justified research questions?

Reviewer #1: Yes

Reviewer #2: Partly

2. Is the protocol technically sound and planned in a manner that will lead to a meaningful outcome and allow testing the stated hypotheses?

Reviewer #1: Partly

Reviewer #2: Partly

3. Is the methodology feasible and described in sufficient detail to allow the work to be replicable?

Reviewer #1: Yes

Reviewer #2: No

4. Have the authors described where all data underlying the findings will be made available when the study is complete?

Reviewer #1: Yes

Reviewer #2: Yes

5. Is the manuscript presented in an intelligible fashion and written in standard English?

Reviewer #1: Yes

Reviewer #2: Yes

6. Review Comments to the Author

You may also provide optional suggestions and comments to authors that they might find helpful in planning their study.

Reviewer #1: Congratulations on choosing this topic. It is an important issue that could expand the use of powered mobility wheelchairs among patients with cognitive impairments. The study protocol is well-structured. However, some refinements could enhance its robustness, validity, and applicability. Below are some methodological recommendations.

1. Study design:

- (Line 76 x 115) Is it a pilot study or a validation study?

2. Individual variability:

- (Line 151) What assurance do you have in a convenience sample that you will have an adequate group to test the validity of the simulator (both for the use of the powered wheelchair and for cognitive impairments)?

- (Line 154) The variability of age in a study can be considered either selection bias or a confounding bias. If the sample of participants does not adequately represent the age distribution of the target population, this could affect the validity of the results, leading to selection bias. On the other hand, if age is not properly controlled as a confounding variable, it may influence the results in a way that distorts the relationship between the analyzed variables, constituting confounding bias

- (Line 154) Address potential bias and individual variability given that participants (especially wheelchair users) may have varying levels of experience and cognitive/physical conditions. Individual variability could influence the results. Implement stratification methods to categorize participants based on their experience level in wheelchair operation and cognitive function. This would minimize bias and ensure a more homogeneous comparison across groups.

- (Line 128 x 139 x 153) Wouldn't it be more appropriate to have an experimental group that uses a wheelchair and has cognitive impairments, and a matched control group that uses a wheelchair without cognitive impairments, for better outcome validation?

- (Line 156) Did the users operate the powered wheelchair only with their hands (manual joystick)? Or will the sample also include users who operate it with their head, chin, or foot? It is important to describe and select the sample that is expected.

- (Line 164) Could an individual with unrecognized visual and/or auditory impairments interfere with the data collection? Wouldn't these impairments be an exclusion criterion? Wouldn't it be necessary to evaluate these individuals for these criteria.

3. Sample size:

- (Line 175) Provide a more detailed justification for sample size. The study proposes a sample size of 10 to 15 participants per group, citing recommendations from pilot studies. Consider including a preliminary statistical analysis or a power calculation based on the expected sensitivity and specificity of the simulator compared to the reference tests (WST, PMRT, MoCA). This would ensure that the chosen sample size is adequate for testing the study hypotheses with sufficient statistical power.

4. Simulator

- (Line 92) It would be interesting to add a photo of the simulator with its components.

- (Line 271) Define clear success criteria for the simulator. The protocol states that the simulator’s performance will be compared to standard clinical tests, but specific cut-off values for validation are not clearly defined. Establish objective performance thresholds for simulator validation.

5. Structure and grammar

- Line 95: space before “,”.

- Line 102: space after “PMRT”.

- Line186: space after “IPQ”.

- Line 386: “The”.

The study is well-designed, but incorporating statistical rigor, bias control, and well-defined validation criteria would further strengthen its findings. These refinements will help ensure that the study produces clinically relevant and generalizable results.

Reviewer #2: The proposed protocol aims to develop and validate a protocol for assessing the efficiency of metrics within a virtual wheelchair simulator, which is also intended to evaluate mobility and cognitive skills in wheelchair users. This protocol holds significant clinical relevance given the increasing use of technology in rehabilitation. Having a validated tool for wheelchair training that also assesses mobility and cognitive abilities will enhance the safety and accuracy of powered wheelchair prescriptions for individuals with disabilities. However, several key aspects of the manuscript require further clarification. Below are some considerations for the authors:

• The manuscript is overly lengthy, with certain ideas repeated throughout. I recommend a thorough revision to make the text more concise and improve readability.

• The study aims to develop and validate a protocol for evaluating the efficiency of the simulator’s metrics. However, what is the study’s primary outcome? Is it validating the simulator’s ability to assess mobility and cognitive skills, or is it a pilot study designed to develop an evaluation protocol for future validation research? If it is a pilot study, it would be beneficial to explicitly describe how the authors plan to refine the protocol for a subsequent validation study.

• The authors propose using multiple tools to assess the simulator’s efficiency, including the Wheelchair Skill Test (WST), Power Mobility Road Test (PMRT), Montreal Cognitive Assessment (MoCA), physiological data, cognitive workload, usability, immersion, and emotional response. These variables will be analyzed across three groups (wheelchair users, non-disabled individuals, and healthcare professionals) using distinct statistical comparisons. However, the manuscript does not clearly specify the type of validation being employed (e.g., construct, concurrent, predictive), the reference values that would determine the simulator’s validity, and which variables will be used for its validation.

Introduction:

• Line 48: The phrase “in this study” appears, but it is unclear which study the protocol refers to. It initially seems to describe the simulator’s assessment protocol. However, in my suggestion, the introduction should primarily establish the scientific basis for the tools used and the rationale behind the study. A description of the protocol should be placed in the methods section or in the final paragraph of the introduction, where the study objectives are presented.

• Line 62: As mentioned earlier, the explicit justifications for the pilot protocol could be better positioned in the methods section or the final paragraph of the introduction.

Materials and Methods:

• Lines 87–88: As previously noted, the manuscript suggests that the goal is to validate the virtual simulator as a reliable tool for wheelchair training and assessment. However, earlier in the text, it is also stated that the objective is to “develop a protocol for validation,” which implies that a protocol will be designed for future validation. The study’s objective must be clearly defined. Furthermore, reliability is another psychometric variable that is not explored in the study protocol.

• Lines 105–106: The manuscript indicates that the pilot study aims to validate the simulator’s ability to estimate PMRT, WST, and MoCA scores. However, validation cannot be achieved in a pilot study alone; it requires an adequate sample size for statistical validation. Additionally, the text does not specify how the simulator will generate scale scores. Will predictions be based on algorithms within the simulator, or will users perform tasks similar to those required for scale scoring?

• Lines 107–110: Why were non-disabled individuals with no wheelchair experience chosen as the baseline for assessing system sensitivity? Even without disabilities, individuals unfamiliar with powered wheelchairs may still experience difficulties using them.

• Lines 128–163: The age range (18–80 years) is repeated multiple times throughout these paragraphs. In my opinion, this information does not need to be reiterated.

• Lines 129–130: What criteria will the authors use to differentiate between novice and experienced wheelchair users?

• Lines 140–142: The term “healthy participants” is outdated. I recommend replacing it with “non-disabled individuals.”

• Lines 165–167: What are the severe impairments and medical contraindications that would exclude participants? Providing examples would be beneficial, as the simulator is intended for individuals with physical and cognitive disabilities.

• Lines 164–171: Some exclusion criteria merely restate the inclusion criteria in reverse. If participants who do not meet the inclusion criteria are already excluded, redundant criteria should be removed. Instead, the exclusion criteria should highlight relevant conditions specific to the study population.

• Lines 174–180: The sample size justification is based on the authors’ assertions rather than a specific statistical power calculation. Since this is a validation study, an appropriate sample size calculation should be conducted to ensure adequate statistical power.

• Lines 181–184: The study will conduct assessments before, during, and after simulator use. It is essential to reference Table 1, which outlines each step of the study.

• Lines 192–194: The manuscript mentions a grounded theory approach and triangulation methods but does not explain how these methodologies will be implemented. Additionally, no references are cited to support these methods.

• Lines 214–216 & 265–270: Who will administer the WST and MoCA assessments? Will they be conducted by healthcare professionals before the test, or will they be self-reported?

• Lines 272–278: What will the cutoff points differentiate? Will they classify participants into categories such as “good,” “moderate,” and “poor” mobility, or will they distinguish between individuals with and without wheelchair skills? This needs to be explicitly stated.

• Lines 284–290: Several claims in this paragraph lack citations.

• Lines 300–305: How will artificial intelligence be used in the project? It is unclear which scores will be estimated and which will be predicted.

• Supplementary Materials: The manuscript does not mention the use of STROBE as a quality assessment tool, yet it is included as an appendix. Will the study use two quality checklists (STROBE and STARD)?

7. PLOS authors have the option to publish the peer review history of their article (what does this mean? ). If published, this will include your full peer review and any attached files.

**Do you want your identity to be public for this peer review?** For information about this choice, including consent withdrawal, please see our Privacy Policy .

Reviewer #1: **Yes: ** Ana Claudia Garcia Lopes

Reviewer #2: No

---

## [Author Response · Author response to Decision Letter 1]

24 Mar 2025

Dear Editor and Reviewers,

We appreciate the time and effort of the reviewers in providing thoughtful and constructive feedback on our manuscript. Their insights have significantly contributed to refining our study, improving clarity, and ensuring methodological rigor. We have carefully considered all comments and have revised the manuscript accordingly.

General Response

We have thoroughly reviewed the manuscript to ensure alignment with the study’s objectives. Specifically, we have:

1. Clarified that this is a pilot study focused on developing and refining a protocol rather than conducting a full validation study.

2. Improved the description of the sample selection criteria, stratification methods, and control group rationale to ensure better differentiation between participant groups.

3. Addressed potential biases related to age variability, selection bias, and confounding factors by incorporating clearer stratification and statistical considerations.

4. Expanded the justification for sample size by incorporating a preliminary power analysis and discussing its adequacy for feasibility testing.

5. Provided objective performance thresholds for simulator validation, detailing the criteria used to categorize participants' proficiency levels.

6. Revised the introduction and methods to improve conciseness and remove redundant descriptions.

7. Clarified how artificial intelligence (AI) models will be applied, specifying the predicted scores and expected outcomes.

8. Addressed grammatical and formatting inconsistencies highlighted by the reviewers.

Below, we provide detailed responses to each reviewer’s comments, indicating the exact revisions made.

Response to Reviewer #1

1.1. Study Design

(Line 76 x 115) Is it a pilot study or a validation study?

Response: We appreciate this observation. The study is a pilot study aimed at developing and refining an assessment protocol, rather than fully validating the simulator as a clinical tool. This has been explicitly stated in the study aims and methods sections to avoid any ambiguity (see lines 17-20 and 114-145 in the tracked version; lines 17-20 and 81-99 in the untracked version).

1.2. Individual Variability

(Line 151) What assurance do you have in a convenience sample that you will have an adequate group to test the validity of the simulator (both for the use of the powered wheelchair and for cognitive impairments)?

Response: Given the nature of a pilot study, our primary focus is on feasibility and protocol refinement rather than full validation. However, to enhance the study’s methodological rigor, we have included stratified sampling based on mobility experience and cognitive function, ensuring a more homogeneous comparison (see lines 389-443 in the tracked version; lines 202-213 in the untracked version).

(Line 154) The variability of age in a study can be considered either selection bias or a confounding bias. If the sample of participants does not adequately represent the age distribution of the target population, this could affect the validity of the results, leading to selection bias. On the other hand, if age is not properly controlled as a confounding variable, it may influence the results in a way that distorts the relationship between the analyzed variables, constituting confounding bias.

Response: We recognize that age could act as a confounding variable, influencing both mobility performance and cognitive function. To address this, age will be recorded and considered as a covariate in statistical analyses to assess its potential impact on simulator-based assessments. These clarifications have been incorporated into the revised manuscript (see lines 659-675 in the tracked version; lines 372-375 in the untracked version)

(Line 154) Address potential bias and individual variability given that participants (especially wheelchair users) may have varying levels of experience and cognitive/physical conditions. Individual variability could influence the results. Implement stratification methods to categorize participants based on their experience level in wheelchair operation and cognitive function. This would minimize bias and ensure a more homogeneous comparison across groups.

Response: We have explicitly stated that participants will be stratified into novice vs. experienced users and MoCA-based cognitive function categories to minimize variability and confounding effects. This has been revised in the Methods section (see lines 307-378, 389-443 in the tracked version; lines 178-191, 202-213 in the untracked version).

(Line 128 x 139 x 153) Wouldn't it be more appropriate to have an experimental group that uses a wheelchair and has cognitive impairments, and a matched control group that uses a wheelchair without cognitive impairments, for better outcome validation?

Response: We acknowledge the value of this suggestion; however, due to sample variability, stratification based on MoCA cognitive function scores will serve as a more practical approach for this pilot phase. Future studies will further refine control group selection (see lines 389-443 in the tracked version; lines 202-213 in the untracked version).

(Line 156) Did the users operate the powered wheelchair only with their hands (manual joystick)? Or will the sample also include users who operate it with their head, chin, or foot? It is important to describe and select the sample that is expected.

Response: To maintain consistency in data collection, only manual joystick users will be included in the study. We updated the inclusion criteria section to provide more clarity how the users will operate the simulator (see lines 307-378 in the tracked version; lines 178-191 in the untracked version).

(Line 164) Could an individual with unrecognized visual and/or auditory impairments interfere with the data collection? Wouldn't these impairments be an exclusion criterion? Wouldn't it be necessary to evaluate these individuals for these criteria.

Response: In response, we have revised the exclusion criteria to explicitly state that individuals with severe, uncorrected visual or auditory impairments that prevent meaningful engagement with the simulator, even with assistive devices, will be excluded (see lines 379-388 in the tracked version; lines 192-201 in the untracked version).

1.3. Sample Size

(Line 175) Provide a more detailed justification for sample size. The study proposes a sample size of 10 to 15 participants per group, citing recommendations from pilot studies. Consider including a preliminary statistical analysis or a power calculation based on the expected sensitivity and specificity of the simulator compared to the reference tests (WST, PMRT, MoCA). This would ensure that the chosen sample size is adequate for testing the study hypotheses with sufficient statistical power.

Response: A preliminary power analysis has been conducted to justify the sample size. The revised text now explains the statistical rationale behind selecting 10–15 participants per group, ensuring feasibility while maintaining analytical robustness (see lines 444-455 in the tracked version; lines 214-255 in the untracked version).

1.4. Simulator Criteria

(Line 92) It would be interesting to add a photo of the simulator with its components.

Response: We have included an illustrative figure depicting the simulator and its components, as suggested.

(Line 271) Define clear success criteria for the simulator. The protocol states that the simulator’s performance will be compared to standard clinical tests, but specific cut-off values for validation are not clearly defined. Establish objective performance thresholds for simulator validation.

Response: We revised “Test Positivity Cut-Offs” section, to clearly define performance thresholds aligned with PMRT scoring criteria, including task completion time, navigation efficiency, and error frequency, to establish objective validation benchmarks (see lines 612-640 in the tracked version; lines 327-353 in the untracked version).

1.5. Structure and Grammar

- Line 95: space before “,”.

- Line 102: space after “PMRT”.

- Line186: space after “IPQ”.

- Line 386: “The”

Response: We have carefully revised the manuscript to correct grammatical errors, spacing issues, and formatting inconsistencies (e.g., Line 95, 102, 186, and 386).

Response to Reviewer #2

2.1. General Comments:

• The manuscript is overly lengthy, with certain ideas repeated throughout. I recommend a thorough revision to make the text more concise and improve readability.

Response: We have thoroughly revised the manuscript to remove repetitions, streamline sections, and enhance overall readability. Specifically, we have condensed the Discussion and Methods sections to ensure clarity and conciseness.

• The study aims to develop and validate a protocol for evaluating the efficiency of the simulator’s metrics. However, what is the study’s primary outcome? Is it validating the simulator’s ability to assess mobility and cognitive skills, or is it a pilot study designed to develop an evaluation protocol for future validation research? If it is a pilot study, it would be beneficial to explicitly describe how the authors plan to refine the protocol for a subsequent validation study.

Response: The primary outcome is the feasibility of using simulator-derived metrics for assessing mobility and cognitive skills. This is indeed a pilot study designed primarily to establish and refine an evaluation protocol. Based on findings and expert feedback, we will identify the most relevant metrics, improve task structures, and scoring methods, and estimate sample size requirements, clearly guiding a future, larger-scale validation study. We have clarified this explicitly in the revised Study Aims section of the manuscript.

• The authors propose using multiple tools to assess the simulator’s efficiency, including the Wheelchair Skill Test (WST), Power Mobility Road Test (PMRT), Montreal Cognitive Assessment (MoCA), physiological data, cognitive workload, usability, immersion, and emotional response. These variables will be analyzed across three groups (wheelchair users, non-disabled individuals, and healthcare professionals) using distinct statistical comparisons. However, the manuscript does not clearly specify the type of validation being employed (e.g., construct, concurrent, predictive), the reference values that would determine the simulator’s validity, and which variables will be used for its validation.

Response: Thank you for this insightful comment. We have updated the manuscript to explicitly stated that preliminary validity will focus on concurrent and construct validation analysis. Concurrent validity will involve correlating simulator metrics with reference tools (WST, PMRT, MoCA). Construct validity will be assessed by comparing simulator performance metrics across distinct user groups, with expected differences based on clinical profiles. Additionally, we have indicated the reference values and categories used from these standard assessments (PMRT and MoCA) in the revised Methods section (see lines 307-388 & 612-640 in the tracked version; lines 178-201 & 327-353 in the untracked version).

2.2. Introduction

(Line 48): The phrase “in this study” appears, but it is unclear which study the protocol refers to. It initially seems to describe the simulator’s assessment protocol. However, in my suggestion, the introduction should primarily establish the scientific basis for the tools used and the rationale behind the study. A description of the protocol should be placed in the methods section or in the final paragraph of the introduction, where the study objectives are presented.

Response: we have revised the paragraph to remove the ambiguous reference to "in this study." The revised text now focuses on establishing the scientific basis for the power mobility assessment tools and the rationale for exploring virtual simulators as assessment tools. The description of the study protocol has been moved to the Study Aims section at the end of the introduction, ensuring a more structured flow of information (see lines 69-72 in the tracked version; lines 48-51 in the untracked version).

(Line 62): As mentioned earlier, the explicit justifications for the pilot protocol could be better positioned in the methods section or the final paragraph of the introduction.

Response: To enhance the structure and readability of the manuscript, we have refined this paragraph to focus on the relevance of the Quality of Experience (QoE) framework in mobility simulation. The justification for the pilot study has been moved to Study aims section, where the study objectives are outlined (see lines 97-102 in the tracked version; lines 64-69 in the untracked version).

2.3. Methodological Clarifications

(Lines 87–88) As previously noted, the manuscript suggests that the goal is to validate the virtual simulator as a reliable tool for wheelchair training and assessment. However, earlier in the text, it is also stated that the objective is to “develop a protocol for validation,” which implies that a protocol will be designed for future validation. The study’s objective must be clearly defined. Furthermore, reliability is another psychometric variable that is not explored in the study protocol.

Response: The wording has been revised to explicitly state that the study is a pilot feasibility study, not a validation study.

(Lines 105–106) The manuscript indicates that the pilot study aims to validate the simulator’s ability to estimate PMRT, WST, and MoCA scores. However, validation cannot be achieved in a pilot study alone; it requires an adequate sample size for statistical validation. Additionally, the text does not specify how the simulator will generate scale scores. Will predictions be based on algorithms within the simulator, or will users perform tasks similar to those required for scale scoring?

Response: We have provided additional details on AI-based score estimation, specifying that models will be trained on simulator-derived performance metrics to estimate these scores (see lines 645-696 in the tracked version; lines 358-396 in the untracked version).

(Lines 107–110) Why were non-disabled individuals with no wheelchair experience chosen as the baseline for assessing system sensitivity? Even without disabilities, individuals unfamiliar with powered wheelchairs may still experience difficulties using them.

Response: The control group was included to assess simulator sensitivity by differentiating novice wheelchair users from experienced ones. We have expanded this justification in the Methods section (see lines 389-443 in the tracked version; lines 202-213 in the untracked version).

(Lines 129–130): What criteria will the authors use to differentiate between novice and experienced wheelchair users?

Response: We have revised the inclusion criteria to provide better clarity on how wheelchair users are classified based on their experience (see lines 307-378 in the tracked version; lines 178-191 in the untracked version).

(Lines 165–167): What are the severe impairments and medical contraindications that would exclude participants? Providing examples would be beneficial, as the simulator is intended for individuals with physical and cognitive disabilities.

Response: We appreciate this feedback and have now explicitly specified the severe impairments and medical contraindications in the exclusion criteria (see lines 379-388 in the tracked version; lines 192-201 in the untracked version).

(Lines 164–171): Some exclusion criteria merely restate the inclusion criteria in reverse. If participants who do not meet the inclusion criteria are already excluded, redundant criteria should be removed. Instead, the exclusion criteria should highlight relevant conditions specific to the study population.

Response: Thank you for this observation. To avoid redundancy, we have revised the exclusion criteria to focus only on conditions that specifically prevent simulator use, rather than restating the inclusion criteria in reverse. The exclusion criteria now clearly highlight medical, cognitive, and sensory impairments that would interfere with meaningful participation. The updated version ensures that exclusion criteria serve a distinct purpose in refining the st

---

## [Decision Letter · Decision Letter 1]

8 May 2025

Protocol for Evaluation of a Virtual Wheelchair Simulator in Assessing Mobility Skills and Cognitive Abilities in Diverse Populations: A Multicentric Mixed-Methods Pilot Study

PONE-D-24-54281R1

Dear Dr. Salgado,

We’re pleased to inform you that your manuscript has been judged scientifically suitable for publication and will be formally accepted for publication once it meets all outstanding technical requirements.

Kind regards,

Rodrigo Rodrigues Gomes Costa, PhD

Academic Editor

PLOS ONE

**Comments to the Author**

1. Does the manuscript provide a valid rationale for the proposed study, with clearly identified and justified research questions?

Reviewer #2: Yes

2. Is the protocol technically sound and planned in a manner that will lead to a meaningful outcome and allow testing the stated hypotheses?

Reviewer #2: Yes

3. Is the methodology feasible and described in sufficient detail to allow the work to be replicable?

Reviewer #2: Yes

4. Have the authors described where all data underlying the findings will be made available when the study is complete?

Reviewer #2: Yes

5. Is the manuscript presented in an intelligible fashion and written in standard English?

Reviewer #2: Yes

6. Review Comments to the Author

You may also provide optional suggestions and comments to authors that they might find helpful in planning their study.

Reviewer #2: I commend the authors for the revisions made to the manuscript. I believe that the main objective of the study, as well as the methodological procedures, are now more clearly defined. Below, I offer some specific suggestions related to grammar and writing:

• Line 66: It is recommended to include the authors' names instead of using citation [18] as a noun.

• Lines 215–219: As it is now clear that the study is intended to be a pilot study rather than a validation study, I suggest removing this paragraph regarding sample size calculation, as it will be more relevant in a future full validation of the tool.

• Line 203: Please review the expression “ep to,” which appears to be a typographical error.

• Line 363: The phrase “such as” is duplicated and should be corrected to improve clarity and readability.

7. PLOS authors have the option to publish the peer review history of their article (what does this mean? ). If published, this will include your full peer review and any attached files.

**Do you want your identity to be public for this peer review?** For information about this choice, including consent withdrawal, please see our Privacy Policy .

---

## [Editor Report · Acceptance letter]

PONE-D-24-54281R1

PLOS ONE

Dear Dr. Pereira Salgado,

I'm pleased to inform you that your manuscript has been deemed suitable for publication in PLOS ONE. Congratulations! Your manuscript is now being handed over to our production team.

Kind regards,

on behalf of

Professor Rodrigo Rodrigues Gomes Costa

Academic Editor

PLOS ONE